# From small to large language models: How much confidence can we have?

## Abstract

In this paper, we provide novel insights into information-based, consistency-based and self-verbalized uncertainty quantification (UQ) for multi-label text classification across a range of recent language models on a new unsaturated benchmark of medical device adverse event reports with interdependent labels. We compare more than twenty encoder- and decoder-only language models across three paradigms: discriminative fine-tuning, generative fine-tuning, and few-shot in-context prompting (instruction-tuned and reasoning variants, local and API-accessible). UQ is performed using token-information measures, consistency under stochastic generation, and self-verbalized confidence, with utility assessed via selective prediction. We provide practical guidance on model selection, when fine-tuning is preferable to prompting, and which UQ signals are most effective for routing and human-in-the-loop triage. Our results reveal trade-offs across model types. Discriminatively fine-tuned decoders achieve the strongest head–tail accuracy while still offering solid uncertainty quantification (UQ). In contrast, generative fine-tuning provides the most reliable UQ overall. Reasoning models improve performance on extreme-tail labels but yield weak UQ. Finally, self-verbalized confidence proves unreliable as an indicator of model certainty.

## 1 Introduction

Large language models (LLMs) are increasingly being applied in domains such as education, academia, and industry. While their use is highly efficient and generally accessible to a broad audience, the development of these systems remains concentrated within a relatively small group of organizations. This imbalance has sparked interest in smaller language models, which offer users and developers more flexibility to adapt performance and to explore additional aspects of trustworthiness, such as uncertainty quantification (UQ). Although both small and large language models have already demonstrated remarkable levels of performance, accuracy alone is often not sufficient to ensure broad adoption. Trust-related criteria are equally important, especially in high-stakes contexts like healthcare, where the need for human oversight becomes paramount (Shneiderman, 2020). Nevertheless, despite their relevance, considerations of trustworthiness are still widely underexplored and often overlooked in the existing literature. Many of the aforementioned aspects depend further on the specific use case. Multi-label classification, though highly relevant in practice, remains both understudied and often overlooked in the literature. It is central to a wide range of applications, including document categorization, clinical coding, and email filtering. Language models could play an important role in tackling such tasks, provided they can be applied in a trustworthy manner. However, the rapid pace of model development and availability presents practitioners with challenging choices: which model (e.g., ModernBERT, LLaMA, etc.) is most suitable? If necessary, which learning paradigm should be adopted (e.g., fine-tuning, in-context learning, etc.)? And how reliable are the resulting predictions? Effective UQ can provide a pathway toward more reliable and trustworthy systems (Ojha et al., 2025; Liu et al., 2024). However, as methods differ substantially between discriminative and generative language models, it is essential to understand and select the appropriate approach for each model and learning paradigm. For smaller, discriminative models, entropy-based measures of uncertainty are relatively well studied and better understood (Baur et al., 2025). In contrast, for large generative models, uncertainty quantification becomes more challenging and complex: token-level uncertainty, consistency across multiple generations, and self-verbalized confidence all introduce new difficulties and might be miscalibrated. As a result, determining the

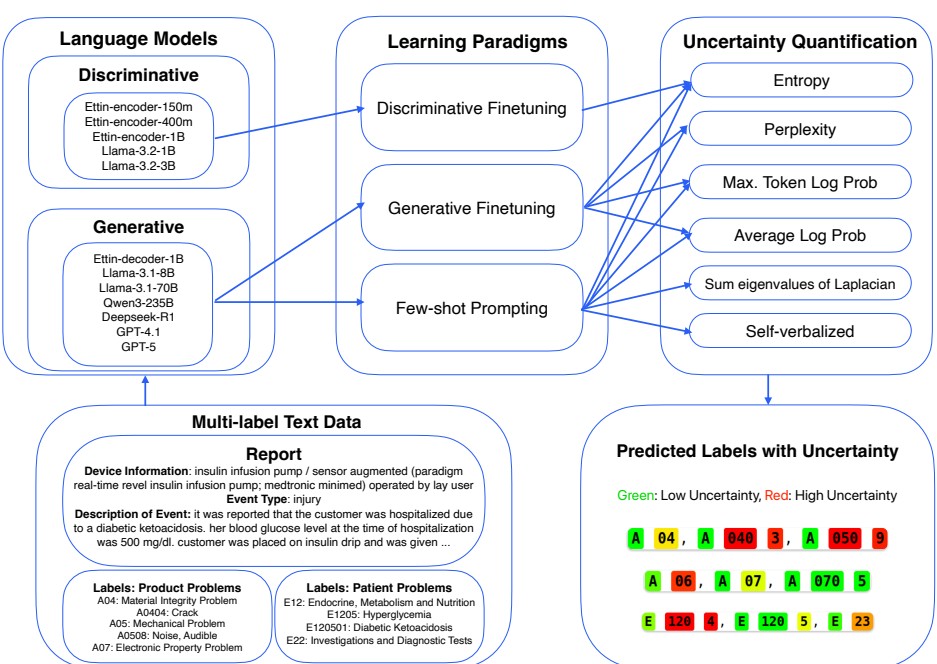

**Figure 1:** Schematic overview of the experimental setup, with an exemplary text report, hierarchical labels (product/patient problems; hierarchy indicated by the digits; e.g., label 'A0404' is a child of 'A04'), models and learning paradigms, uncertainty quantification methods, and predicted labels with (color coded) uncertainty. We finally assess the predictive performance and quality of the uncertainty quantification (not depicted).

most effective UQ strategy – for purposes such as sample routing and efficient human-in-the-loop triage – remains an open research question.

**Our contribution.** To avoid reliance on saturated benchmarks and potential pre-training exposure, we constructed a new dataset of medical device event reports, serving as a prototypical use case for exploring multi-label classification not only in terms of predictive performance but also across various aspects of uncertainty quantification (Figure 1), offering insights into trustworthiness and reliability across over 20 popular language models and different UQ methods. In particular, we (1) create a new unsaturated multi-label text classification dataset with a skewed distribution of hierarchical and thus interdependent labels; (2) fine-tune over twenty encoder- and decoder-only models in discriminative and generative settings, and compare them with zero-/few-shot prompting of locally-hosted decoders and API-accessible models with the proposed dataset; (3) systematically study multiple UQ approaches tailored to these settings – including information-, consistency- and self-verbalized based uncertainty – and study their quality (and thus ultimately, their utility for sample routing or triage).

Overall, our results provide practical guidance on language model choice, on when fine-tuning is preferable to in-context learning, and on how to select and calibrate UQ methods, such that they are suited for routing ambiguous samples to stronger but more expensive models or flagging them for secondary human review.

## 2 RELATED WORK

### 2.1 MULTI-LABEL TEXT CLASSIFICATION

Early approaches to multi-label text classification relied on classical models such as Naïve Bayes, support vector machines (Wang & Manning, 2012), or recurrent neural networks with mixed results.

With the advent of transformer-based language models (Vaswani et al., 2017), both encoder-only (Huang et al., 2021) and decoder-only (Ma et al., 2025; Galke et al., 2025) architectures have been applied to this task. Encoder-only models are typically equipped with a linear classification head and discriminatively fine-tuned on multi-labeled text, usually with a variant of the binary cross-entropy loss. Decoder-only models are often fine-tuned generatively—by producing label tokens via a softmax decoder head — and can also be adapted for discriminative training with a classification head; however, controlled comparisons between generative and discriminative fine-tuning for decoder models remain scarce. Head-to-head comparisons between encoder- and decoder-only models under matched conditions are also rare, largely because publicly available suites have historically lacked paired models with comparable parameter counts and training data. Weller et al. (2025) address this by releasing a suite of matched encoder-only and decoder-only models ('Ettin') trained on the ModernBERT architecture (Warner et al., 2024b) across model sizes (from 17m parameters to 1b parameters), but they do not evaluate on multi-label classification, leaving this comparison open.

Decoder-only LLMs support zero-shot prompting – performing inference without using any labeled data – and few-shot prompting via in-context learning, i.e., conditioning on a small set of (random or carefully selected) labeled exemplars without updating model weights. This has been compared with fine-tuning on a variety of tasks where medical expertise is required, with differing results. Nori et al. (2023) and Maharjan et al. (2024) found that in-context learning outperforms fine-tuning on question-answering. In contrast, Labrak et al. (2024) found that fine-tuning surpassed few-shot prompting on question-answering. Across broader benchmarks, the relative advantage of fine-tuning versus prompting varies by task, data regime, and model (Chen et al., 2025; Wu et al., 2025). For multi-label classification specifically, performance is highly sensitive to prompt design, label semantics, output constraints, and thresholding strategies, motivating controlled comparisons.

Public benchmarks for multi-label text classification span diverse domains, including newswire ('Reuters-21578', Lewis, 1997), scientific literature (e.g., 'Arxiv Academic Paper Dataset', Yang et al., 2018; 'BioCreative LitCovid', Chen et al., 2022; 'ACL Anthology', Schopf et al., 2023; 'MFHAD2', Fallah et al., 2022), finance-related user posts ('Money.StackExchange', Maia et al., 2021), patents ('Patent_CIRCA_45k', Tang et al., 2020)), clinical notes for ICD coding ('MIMIC-III', Johnson et al., 2016; Mullenbach et al., 2018), legislative documents ('EuroVoc', Steinberger et al., 2012, Boella et al., 2013; 'EURLEX57K', Chalkidis et al., 2019; 'KEVLAR', Bocchi et al., 2024), and more. Most of these are scraped from online repositories and rely on distant or weak supervision (e.g., tags/metadata as labels), often with limited documentation, which introduces label noise and hampers reproducibility. Multiple sources, preprocessing choices, copyright limitations, and dataset versions further complicate fair comparisons. Heavy reuse has also lead to benchmark saturation via overfitting and implicit adaptation. In addition, overlap between benchmark content and LLM pre-training corpora raises contamination concerns that can inflate zero-/few-shot performance. These challenges motivate periodically introducing better-documented, decontaminated benchmarks with standardized splits to assess generalization.

## 2.2 UNCERTAINTY QUANTIFICATION

For quantifying uncertainty in machine learning models, numerous approaches have been proposed. Confidence calibration for imbalanced data has been reviewed by Dong et al. (2025). Token- or information-based uncertainty metrics, which in various forms rely on the log-probabilities of an LLM's outputs, have been surveyed and formalized by Shorinwa et al. (2025); Fomicheva et al. (2020). Among those of interest are: entropy of the top-$n$ log-probabilities, perplexity, maximum token log-probability, and the average token log-probability. Consistency-based metrics, which estimate uncertainty by measuring the variability of a model's outputs under input perturbations, prompt variations, or model stochasticity, have been widely used for LLMs as a way to capture epistemic uncertainty Xiao et al. (2025). Lin et al. (2023) suggested an effective approach for deriving a consistency-based uncertainty score through the sum of eigenvalues of a Laplacian graph. Combining information-based and consistency-based metrics has been proposed by Vashurin et al. (2025) as an effective strategy to improve uncertainty estimation, motivating our approach of multiplicatively integrating token-level uncertainty with consistency measures. Finally, self-verbalized uncertainty, in which a model outputs a confidence score along with its prediction, has been explored by Tian et al. (2023) and Harsha Tanneru et al. (2024), offering a complementary approach to directly capture model-reported uncertainty, especially in cases where log-probabilities are not available.

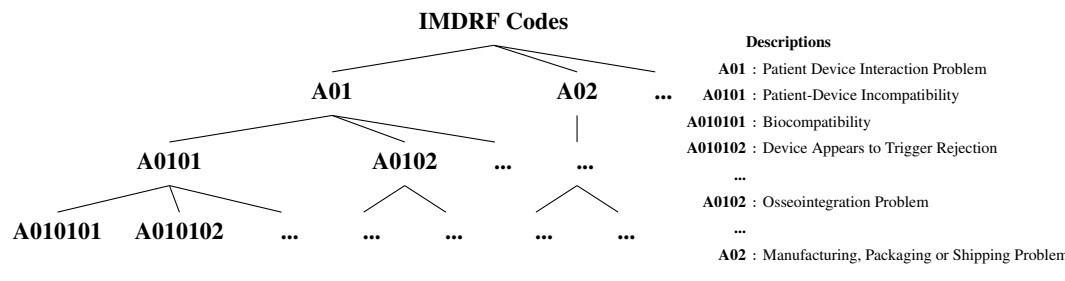

**Figure 2:** IMDRF codes refer to product (device) and patient problems (defined in the different annexes A, B, C, etc. of the IMDRF terminology) and comprise the hierarchical labels of the dataset. IMDRF code 'A010101' is a child of 'A0101' and grandchild of 'A01' in the code hierarchy (a graph not a tree).

# 3 DATA – FDA MEDICAL DEVICE ADVERSE EVENT REPORTS

The US Food and Drug Administration (FDA) publishes quarterly reports of medical device adverse events at `https://open.fda.gov` alongside expertly annotated codes capturing the associated problems. We used this to curate a large-scale text classification dataset comprising approximately 5.5 million medical device adverse event reports, each annotated with hierarchical multi-labels, cf. Figure 2. Each report contains a narrative description of the event, metadata, and human-assigned labels that detail relevant product and patient problems.

## 3.1 DATASET CREATION

We collect quarterly published (zipped JSON) files spanning 2015 through mid-2025, excluding reports before 2015 due to the absence of necessary product problem labels. Data from 2015–2023 are used for training, while test data comprises reports from July 2024 to June 2025, cf. Figure A.1 for a small overview of the topics addressed in the reports. Reports from the first half of 2024 form the validation split. For each report, we concatenate the narrative 'description of event or problem', harmonize all letter cases, and extract relevant metadata (including event type – for example, malfunction, injury or death – and device information such as generic name, brand name, manufacturer, and operator). Each sample is assigned a unique identifier and a timestamp denoting the report date.

## 3.2 DATASET LABELS

Labels are mapped from FDA product and patient problem terms to the International Medical Device Regulators terminology (IMDRF, 2025), which provides hierarchical codes for each product and patient problem (see Figure 2). Samples missing IMDRF codes for one or more terms are excluded. IMDRF adoption is nearly complete by 2021, but between 2015 and 2020 only about 16–20% of the samples can be assigned exclusively to IMDRF codes and are retained. To leverage the IMDRF hierarchy, we upward-propagate each code to include all ancestor codes, as determined from official IMDRF annexes. This approach recognizes that FDA annotators typically use the most specific (leaf) codes and ensures that models are not penalized for predicting valid parent codes.

Although the IMDRF label hierarchy is often tree-structured (e.g., 'A010101' is a child of 'A0101' and grandchild of 'A01'), it is technically a graph due to occasional cross-links (e.g., 'E172001' is a descendant in multiple branches). Label mapping and upward propagation are performed separately for product and patient problems, yielding two sets of hierarchical labels per report. We flatten the resulting code sets and treat the union of upward-propagated IMDRF codes as target labels. Generative models can produce these sets directly, while discriminative models use a binary multi-label vector over the IMDRF codes present in the train split.

## 3.3 DATASET PRE-PROCESSING

IMDRF revises its label taxonomy annually, adding new terms and retiring others. To avoid introducing unseen labels into the test set, we freeze the label taxonomy as of December 2023 and discard

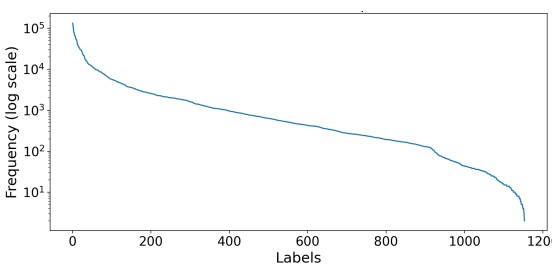

**Figure 3:** Distribution of label frequencies (in the combined train, validation, and truncated test set) on a logarithmic scale, showing that some labels occur very often in the dataset, while others are extremely rare.

labels introduced thereafter. We also remove extremely rare labels (fewer than 5 instances). These steps yield a final label set of 1154 labels. To reduce redundancy, we deduplicate reports based on the event description, retaining only the first occurrence, and thus reducing the dataset to 5.5 million IMDRF-compliant samples (46.4% of the original 11.9 million).

We further downsample using HDBSCAN (McInnes et al., 2017) on event descriptions, selecting cluster representatives and retaining a high proportion of rarer labels; all events labeled death are included, yielding a dataset of 488,273 samples (train: 298,825; validation: 71,271; test: 118,177). To limit inference costs, we further downsample the test set to a representative subset of 10288 samples. This 'truncated test set' set contains all of the 1154 labels at least 5 times. After tokenising with the 'cl100k_base' tokenizer (OpenAI, 2022), samples in the truncated test set average around 370 tokens with an average of 8.79 labels. On all splits, labels follow a long tail distribution; certain labels occur frequently, while others are rare (see Figure 3). In summary, our pre-processing pipeline transforms raw FDA device event data into a IMDRF terminology compliant, hierarchically labeled, deduplicated dataset tailored for a challenging multi-label classification task.

## 4 EXPERIMENTAL SETUP

### 4.1 FINE-TUNING AND IN-CONTEXT LEARNING

**Discriminative training.** We fine-tune the recent Ettin suite of models (150M, 400M, 1B; Weller et al., 2025), which are based on the ModernBERT (Warner et al., 2024a) architecture. We also fine-tune Llama 3.2-1B, 3.2-3B, and 3.1-8B (Grattafiori et al., 2024) with a classification head in the final layer. All models are trained with a context length of 512 for 20 epochs each and hyperparameters (mainly learning rate and max. gradient norm.), are tuned on the validation set

**Generative training.** We fine-tune Llama 3.2-1B, 3.2-3B, 3.1-8B and 3.1-70B and Ettin decoder variants (400M, 1B), where the model is trained to generate class labels as tokens. Here, all models are trained for 4 epochs. An improvement for a larger number of epochs was not observed. We compare full fine-tuning with LoRA (Hu et al., 2022) for selected models.

Detailed hyperparameters for training are listed in the appendix section A.3.

**Few-shot prompting and in-context learning.** We host Llama 3.2-3B, 3.1-8B, and 3.1-70B, DeepSeek-R1 (37B active, 671B total; DeepSeek-AI et al., 2025), Qwen3 (4B, 30B, 235B; Yang et al., 2025), Kimi K2 (32B activated; 1T total; Kimi-Team et al., 2025), gpt-oss-120b (5.1B active, 117B total; OpenAI et al., 2025), GLM-4.5-Air (12B active, 106B total; GLM-4.5-Team et al., 2025), and Llama-3.3-Nemotron-49B-v1.5 (Bercovich et al., 2025) locally. GPT-4.1 (OpenAI, 2025b) and GPT-5 (OpenAI, 2025a) are accessed via API. Among API-based models, only GPT-4.1 provides – unlike GPT-5 – log probabilities, which are required for our UQ investigations. GPT-5 is run with medium reasoning effort and does not allow modifying temperature.

The prompt comprises the test sample, the task instructions including a list of all 1154 possible labels, and the ten nearest labeled training examples in embedding space to the inference sample. For embedding the texts, we use a pre-trained text embedding model ('bioclinical-modernbert-base-embeddings') (NeuML, 2025). Each retrieved example was truncated to 10000 characters (around 2300 tokens). Unless otherwise stated, all outputs are generated in a greedy decoding setup ($temperature = 0, top\_p = 1, top\_k = -1$). Prompts are available in the appendix A.5.

The knowledge cutoff dates for all the models are discussed in Section A.2 in the appendix.

## 4.2 Methods for uncertainty quantification

**Uncertainty quantification for generative models.** To quantify uncertainty $U$, we derive two complementary measures: an information-based uncertainty score $U_{\text{info}}$ and a consistency-based uncertainty score $U_{\text{cons}}$. Following Vashurin et al. (2025), we combine these scores by multiplication to obtain the total uncertainty score: $U_{\text{total}} = U_{\text{info}} \cdot U_{\text{cons}}$. Here, $U_{\text{info}}$ is computed using token-based uncertainty metrics such as entropy, average log-probability, perplexity, maximum log-probability, and response improbability (described below), while $U_{\text{cons}}$ is computed using the Laplacian of a graph derived from multiple stochastic forward passes (Lin et al., 2024). For large language models (LLMs), we evaluate these token- and information-based uncertainty scores that serve as $U_{\text{info}}$:

- Entropy of Token Distribution: $H_{ij} = -\sum_{w \in D} \pi_{ij}(w) \log \pi_{ij}(w)$
- Average Token Log-Probability: $\text{Avg}(\pi) = -\frac{1}{L_i} \sum_j \log(\pi_{ij})$
- Perplexity: $\text{Perplexity}(\pi) = \exp\left(-\frac{1}{L_i} \sum_j \log(\pi_{ij})\right)$
- Maximum Token Log-Probability: $\text{Max}(\pi) = \max_j \left(-\log(\pi_{ij})\right)$
- Response Improbability: $\text{Improbability} = 1 - \prod_j \pi_{ij}$

The consistency-based score is computed by performing $n = 5$ stochastic forward passes with temperature $t = 1$, calculating the pairwise Jaccard similarity $W$ between predictions, and computing the normalized Laplacian $L$ as $L = I - D^{-1/2} W D^{-1/2}$, where $I$ is the identity matrix and $D$ is the degree matrix ($D = \text{diag}\left(\sum_j W_{ij}\right)$).

The consistency score is then $U_{\text{cons}} = \sum_k \max(0, 1 - \lambda_k)$, with $\lambda_k$ denoting the eigenvalues of $L$.

We also evaluate self-verbalized uncertainty, for which we prompt the model to provide a confidence score $C_i \in [0, 1]$ for each prediction. We then define the self-verbalized uncertainty score as $U_{\text{self}} = 1 - C_i$, which can also serve as an independent total uncertainty score for the prediction. This formulation ensures that high confidence corresponds to low uncertainty, and vice versa.

**Uncertainty quantification for discriminative models.** For discriminative models in a multi-label setting, uncertainty is derived from the model's output probability vector. Since each label is binary and independent of the others, we first compute the binary entropy for each predicted label:

$$\mathbb{H}(\pi_c) = -\big[\pi_c \log(\pi_c) + (1 - \pi_c) \log(1 - \pi_c)\big],$$

where $\pi_c$ is the predicted probability for the positive class (label 1) of class $c$. The information-based uncertainty for a sample is then the vector of these entropies across all labels:

$$U_{\text{info}} = \big(\mathbb{H}(\pi_1), \mathbb{H}(\pi_2), \ldots, \mathbb{H}(\pi_C)\big).$$

## 4.3 Evaluation of uncertainty scores

To evaluate the quality of uncertainty scores (either $U_{\text{total}}$ or $U_{\text{info}}$), we perform selective prediction. In this setup, a model's outputs are rejected based on their uncertainty scores. We then assess if choosing to reject an output improves the overall prediction metric.

**Prediction rejection rate (PRR).** PRR quantifies how well an uncertainty score $U$ identifies unreliable predictions. It is computed as:

$$\text{PRR} = \frac{\text{AUC}_{\text{uncertainty}} - \text{AUC}_{\text{random}}}{\text{AUC}_{\text{oracle}} - \text{AUC}_{\text{random}}},$$

where $\text{AUC}_{\text{uncertainty}}$ is the area under the curve (AUC) obtained by sorting samples by $U$ (highest uncertainty first) and iteratively rejecting the top $N\%$ of samples, computing the average Jaccard score after each rejection threshold. $\text{AUC}_{\text{oracle}}$ is the AUC obtained by sorting samples by the Jaccard score (lowest score first) and rejecting iteratively – this gives us a clear upper-bound to compare against for a given rejection threshold. $\text{AUC}_{\text{random}}$ is the AUC obtained by randomly rejecting $N\%$ of samples at each step. PRR evaluates the effectiveness of $U$ in prioritizing uncertain predictions. A value close to 1 indicates that the uncertainty score performs nearly as well as the oracle, while a PRR near 0 indicates performance no better than random rejection.

**Spearman correlation ($\rho$) of $U$ with correctness.** To further assess the quality of the total uncertainty score $U$, we correlate $U$ with per-sample correctness. We evaluate binary correctness (0/1) for each example–label pair, compute Spearman's $\rho$ between correctness and uncertainty separately for each label, and then report the average $\rho$ across labels. A large negative correlation indicates that higher uncertainty coincides with lower prediction accuracy, confirming that $U$ effectively flags unreliable predictions.

## 5 RESULTS

We evaluate predictive performance (macro F1, Jaccard $J$) and uncertainty quantification (PRR, Spearman $\rho$) across three paradigms: (i) discriminative fine-tuning of encoders/decoders, (ii) generative fine-tuning of decoders, and (iii) 10-shot kNN-based in-context prompting with instruction-tuned or thinking/reasoning models. Labels are grouped by training-set frequency: head ($>1\%$), medium (0.1–1%), tail (0.01–0.1%), and extreme tail ($<0.01\%$). Table 1 summarizes all results.

### 5.1 MULTI-LABEL PREDICTIVE PERFORMANCE

**Fine-tuning.** Llama-3.1-8B-Base yields the highest macro F1 overall (0.54), and leads on head (0.74), medium (0.64), and tail (0.53) classes. Smaller Llamas (3.2-3B/1B) and encoders (Ettin-1B/400m/150m) trail slightly, consistent with model capacity, but Ettin-1B-Encoder remains competitive given its size. Llama-3.1-70B-Base, despite being significantly larger, achieves similar results to Llama-3.1-8B-Base in a generative setting. Notably, fine-tuning similarly-sized decoders in a discriminative setting outperforms fine-tuning in a generative setting. We include ablations on LoRA vs. full fine-tuning (Table A.1) and Base vs. Instruct training (Table A.2) in the appendix.

**Generative prompting.** For prompting setups with 10-shot in-context learning, Qwen3-235B-A22B-Instruct achieves the strongest macro F1 (overall 0.44, extreme tail 0.24), with only GPT-4.1 matching tail performance. Reasoning models excel on the extreme tail (ET): GPT-5 attains the highest ET macro F1 (0.34) and ties the best overall macro F1 (0.54) with Llama-3.1-8B-Base (discriminative). Among open-weight reasoning models, Qwen3-235B-A22B-Thinking is strongest (overall 0.49, tail 0.48, ET 0.33). Gain on rare labels for all prompt-based models comes with a trade-off: head-class performance is consistently below that of the best fine-tuned decoders.

**Overall.** Discriminative fine-tuning of decoders (Llama-3.1-8B-Base) remains best for head–tail accuracy – this approach consistently outperforms generative training. Prompt-based reasoning models (Qwen3-235B, DeepSeek-R1, GPT-5) dominate the extreme tail and can match overall F1 but lag on head classes. Within prompting, reasoning variants consistently outperform their instruct counterparts (see Qwen3 results).

### 5.2 UNCERTAINTY QUANTIFICATION

On PRR, across paradigms, generative fine-tuning yields the strongest UQ, followed by discriminative fine-tuning; prompting instruction models ranks third, and reasoning models are a distant last (Table 3). Figure 4 illustrates PRR curves and how PRR summarizes gating quality (shown for Llama-3.1-70B-Base). Overall, Llama-3.2-3B-Base (generative fine-tuning) achieves the highest PRR (0.70) and $\rho$ (-0.46), while thinking models exhibit low PRR ($\leq 0.36$) and near-zero $\rho$.

For non-reasoning setups that also deliver strong predictive performance, UQ quality is generally good: Qwen3-235B-A22B-Instruct attains PRR 0.54 and Llama-3.1-70B-Instruct reaches PRR 0.61. Importantly, uncertainty quality does not scale monotonically with model size (e.g., within discriminative fine-tuning, Llama-3.2-1B-Base has the strongest PRR, 0.52, while the 8B variant is lower at 0.47), and the best overall PRR is achieved by a mid-sized generatively fine-tuned model rather than the largest models.

Methodologically, Table 2 shows that the choice of UQ method for a given model has only minor impact on PRR. Further, computing $U_{\text{total}}$ yields marginal gains ($\leq 0.03$ PRR) relative to $U_{\text{info}}$ alone, despite incurring roughly 6× compute due to multi-sample generation. Self-verbalized uncertainty performs near chance (PRR = 0), indicating that free-text confidence is not a reliable UQ signal for this task. Finally, closed models that do not expose token-level log-probabilities (e.g., GPT-5) impede UQ and are therefore unsuitable for PRR-based calibration.

**Table 1:** Predictive performance and uncertainty quantification capabilities of different models and learning paradigms on the truncated test set ($n = 10288$). Macro F1 serves as main predictive metric. Head, medium, tail and extreme tail (ET) classes were assessed separately. For discriminative training, thresholds were chosen for each label on the validation set. J refers to the Jaccard score. PRR is the main metric for assessing uncertainty. For generative models, we report PRR for the best uncertainty quantification method (see $U_{\text{info}}$ in 4.3). $\rho$ refers to the Spearman correlation between $U$ and correctness. Up-/downward arrows indicate whether larger ($\uparrow$) or smaller ($\downarrow$) values are better. Bold font marks the best model per paradigm. Scores of the best model overall are underlined.

| Paradigm/model | Macro F1 $\uparrow$ | | | | | J $\uparrow$ | PRR $\uparrow$ | $\rho \downarrow$ |
|---|---|---|---|---|---|---|---|---|
| | **Overall** | **Head** | **Medium** | **Tail** | **ET** | | | |
| *Number of classes* $\rightarrow$ | *1154* | *144* | *481* | *348* | *181* | | | |
| **Discriminative fine-tuning** | | | | | | | | |
| Ettin-150m-Encoder | 0.46 | 0.68 | 0.56 | 0.44 | 0.07 | 0.55 | 0.38 | -0.30 |
| Ettin-400m-Encoder | 0.51 | 0.72 | 0.61 | 0.50 | 0.12 | 0.58 | 0.44 | -0.36 |
| Ettin-1B-Encoder | 0.53 | 0.73 | 0.63 | 0.51 | 0.13 | 0.61 | 0.46 | -0.40 |
| Llama-3.2-1B-Base | 0.51 | 0.71 | 0.60 | 0.48 | **0.14** | 0.58 | **0.52** | **-0.42** |
| Llama-3.2-3B-Base | 0.51 | 0.72 | 0.62 | 0.49 | 0.11 | 0.59 | 0.46 | -0.41 |
| Llama-3.1-8B-Base | 0.54 | 0.74 | 0.64 | 0.53 | 0.12 | 0.62 | 0.47 | -0.40 |
| **Generative fine-tuning** | | | | | | | | |
| Ettin-400m-Decoder | 0.44 | 0.66 | 0.54 | 0.42 | 0.07 | 0.54 | 0.56 | -0.44 |
| Ettin-1B-Decoder | 0.47 | 0.67 | 0.56 | 0.46 | 0.10 | 0.57 | 0.55 | -0.43 |
| Llama-3.2-1B-Base | 0.43 | 0.63 | 0.52 | 0.39 | 0.10 | 0.45 | 0.53 | -0.44 |
| Llama-3.2-3B-Base | 0.48 | 0.67 | 0.57 | 0.46 | 0.12 | 0.58 | **0.70** | **-0.46** |
| Llama-3.1-8B-Base | 0.50 | 0.70 | 0.59 | 0.48 | 0.12 | 0.59 | 0.43 | -0.30 |
| Llama-3.1-70B-Base | **0.53** | **0.73** | **0.62** | **0.51** | **0.16** | **0.61** | 0.39 | -0.27 |
| **Generative prompting – instruct** | | | | | | | | |
| Llama-3.1-8B-Instruct | 0.08 | 0.28 | 0.09 | 0.03 | 0.006 | 0.22 | 0.22 | 0.26 |
| Llama-3.1-70B-Instruct | 0.30 | 0.50 | 0.35 | 0.25 | 0.08 | 0.43 | **0.61** | -0.15 |
| Qwen3-4B-Instruct | 0.29 | 0.49 | 0.35 | 0.25 | 0.09 | 0.41 | 0.50 | -0.27 |
| Qwen3-30B-A3B-Instruct | 0.22 | 0.48 | 0.27 | 0.14 | 0.05 | 0.43 | 0.54 | 0.05 |
| Qwen3-235B-A22B-Instruct | **0.44** | **0.60** | **0.48** | **0.42** | **0.24** | 0.49 | 0.54 | **-0.34** |
| Kimi-K2-Instruct | 0.09 | 0.18 | 0.11 | 0.06 | 0.008 | 0.07 | 0.28 | 0.08 |
| GPT-4.1 | 0.43 | 0.59 | 0.47 | **0.42** | 0.22 | **0.57** | 0.43 | -0.31 |
| **Generative prompting – thinking** | | | | | | | | |
| Llama-3.3-Nemotron-49B-v1.5 | 0.42 | 0.57 | 0.46 | 0.38 | 0.19 | 0.46 | 0.23 | -0.03 |
| Qwen3-4B-Thinking | 0.38 | 0.53 | 0.42 | 0.36 | 0.2 | 0.43 | 0.18 | -0.02 |
| Qwen3-30B-A3B-Thinking | 0.45 | 0.58 | 0.49 | 0.44 | 0.28 | 0.47 | 0.09 | -0.07 |
| Qwen3-235B-A22B-Thinking | 0.49 | 0.62 | 0.52 | 0.48 | 0.33 | 0.48 | **0.36** | **-0.09** |
| GLM-4.5-Air | 0.42 | 0.56 | 0.46 | 0.39 | 0.24 | 0.44 | 0.24 | **-0.09** |
| DeepSeek-R1-0528 | 0.48 | 0.62 | 0.51 | 0.47 | 0.30 | 0.50 | 0.26 | **-0.09** |
| GPT-OSS-120B | 0.40 | 0.57 | 0.45 | 0.38 | 0.15 | 0.45 | 0.06 | -0.0006 |
| GPT-5 | 0.54 | 0.68 | 0.58 | 0.53 | 0.34 | 0.57 | NA | NA |

# 6 DISCUSSION

**Discussion and conclusion.** In this work, we evaluate a range of language models on a newly created unsaturated multi-label text dataset, including different-sized variations of Ettin, LLaMA, Qwen, as well as GLM, Kimi, DeepSeek, and GPT models. Our results show that smaller, task-specific encoder models can achieve state-of-the-art predictive performance while maintaining competitive UQ scores, and additionally offer practitioners flexibility and full control of all model aspects. Large generative fine-tuned models provide slight advantages for underrepresented classes and tend to produce better-calibrated uncertainties. Thinking models in contrast consistently fail to yield a reliable UQ across different model types. Self-verbalized uncertainty in LLMs is highly miscalibrated and should not be used. Prompt-based non-thinking instruct models can produce good but also poor uncertainty scores depending on model type and size, highlighting that fine-tuning is

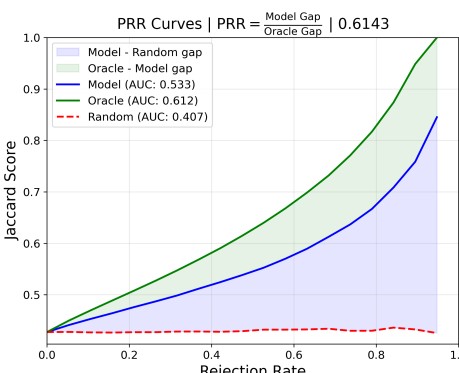

**Figure 4:** Prediction Rejection Rate (PRR) curves for the Jaccard score for Llama 70B. Three curves represent the model performance under different rejection strategies: oracle (green curve), uncertainty (blue curve), and random (red dashes). Shaded areas indicate the performance gaps between the rejection strategies (blue: 'uncertainty gap' between uncertainty and random; green: 'oracle gap' between oracle and uncertainty). The PRR norm is defined as the ratio of the random gap to the uncertainty gap and indicates the suitability of the uncertainty quantification method (and the model).

**Table 2:** Comparison of the different UQ methods: information-based ($U_{\text{info}}$), consistency-based ($U_{\text{cons}}$), and information-and-consistency-based ($U_{\text{total}} = U_{\text{info}} * U_{\text{cons}}$) for two of the best performing (non-thinking) generative models. To compute $U_{\text{cons}}$, the models were made to generate 5 outputs at temperature = 1.

| Model | Method | **PRR**($U_{\text{info}}$) | **PRR**($U_{\text{cons}}$) | **PRR**($U_{\text{total}}$) |
|---|---|---|---|---|
| Llama-3.1-70B-Instruct | Entropy | **0.60** | - | **0.61** |
| | Perplexity | 0.58 | - | 0.60 |
| | Max. Token Log Prob. | 0.57 | - | 0.60 |
| | Avg. Logprob | 0.59 | - | 0.60 |
| | Improbability | 0.58 | - | **0.61** |
| | Σ EigV. Laplacian | - | 0.57 | - |
| | Self-Verbalized | 0.00 | - | 0.36 |
| Qwen3-235B-A22B-Instruct | Entropy | **0.54** | - | **0.57** |
| | Perplexity | 0.53 | - | 0.56 |
| | Max. Token Log Prob. | 0.51 | - | 0.53 |
| | Avg. Logprob | 0.53 | - | 0.54 |
| | Improbability | 0.53 | - | 0.54 |
| | Σ EigV. Laplacian | - | 0.51 | - |
| | Self-Verbalized | 0.01 | - | 0.28 |

**Table 3:** Average PRR scores for different $U_{\text{info}}$ methods for each generative paradigm from Table 1.

| | Entropy | Perplexity | Max. Token Log Prob. | Avg. Logprob. | Improbability |
|---|---|---|---|---|---|
| **Generative fine-tuning** | **0.58** | **0.57** | **0.56** | **0.57** | **0.56** |
| **Generative prompting – instruct** | 0.47 | 0.44 | 0.44 | 0.44 | 0.45 |
| **Generative prompting – thinking** | 0.20 | 0.18 | 0.18 | 0.18 | 0.18 |

generally crucial for obtaining trustworthy uncertainty estimates, for both small discriminative and large generative models. Future work aimed at improving UQ for smaller encoder models could make them the superior choice, combining top-tier predictive performance with trustworthy uncertainty estimates, practical flexibility, and significantly lower computational costs compared to large language models.

**Limitations and outlook.** Our thorough analysis is carried out on a single newly created dataset. In the future, we want to extend the research conducted in this work to other datasets to further solidify the results. Moreover, in future work, other dimensions of trustworthiness, such as explainability, shall be integrated and studied, to increase the benefits for end-users. Finally, based on our observations, training and developing new reasoning models is a promising direction of research. In fact, we will investigate improving UQ for these models while maintaining high predictive performance across both frequent and infrequent classes.

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

# A  APPENDIX

## A.1  DATASET TOPICS

Figure A.1 illustrates the topics addressed in the medical device adverse event reports used for testing.

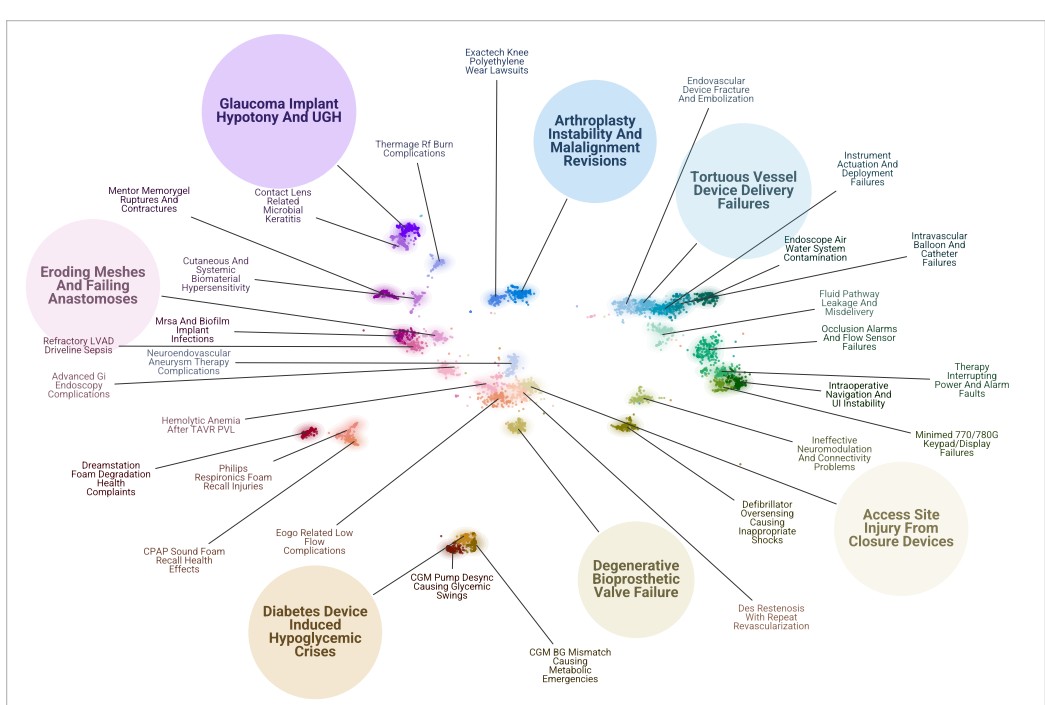

**Figure A.1:** Illustration of 35 most frequent topics in the (truncated) test set. Topics were modeled with UMAP (McInnes et al., 2020) and K-means clustering (Lloyd, 1982), and named with GPT-5.

## A.2  KNOWLEDGE CUTOFF DATES

Reports from July 2024 to June 2025 are our test data (see Section 3). This time span ensures that there are no overlaps with the models' pre-training knowledge cutoff dates for Llama 3.1-3 (December 2023), GPT 4.1 (June 2024) and gpt-oss-120b (June 2024). Reports are published *after* each quarter (reports from July-September 2024 in October 2024 or later). Consequently, our test set should also contain only data released after the GPT-5 knowledge cutoff in September 2024. Ettin (Weller et al., 2025) was pre-trained including on the 'DOLMino mix 1124' dataset (OLMo et al., 2025, which also mention data from September 2024), among other data sources. Llama-3.3-Nemotron-49B-v1.5 used post-training data released in July 2025. We have not found publicly available official information on the knowledge cutoff dates of DeepSeek-R1, Qwen3, and Kimi K2 (released in January, April, and July 2025, respectively).

## A.3  HYPERPARAMETERS

For generative training, we use the AdamW optimizer Loshchilov & Hutter (2019) with cosine learning-rate decay Loshchilov & Hutter (2016), a warmup ratio of 0.1, and a batch size of 64. We tune the learning rate (from 3e-5 to 1e-4) and the maximum gradient norm on the validation set.

For discriminative training, we use AdamW with a cosine scheduler, a fixed batch size of 512, 20 epochs, a warmup ratio of 0.1, and a fixed maximum gradient norm of 0.01; we tune the learning rate in the range [2e-4, 5e-4] on the validation set.

## A.4 ADDITIONAL RESULTS

In a generative setting, full fine-tuning is compared to parameter-efficient fine-tuning using LoRA in Table A.1. For Llama models (1B, 3B), full fine-tuning improves macro-F1 by 0.01. For Ettin models (400M, 1B), the gains are greater, at 0.12 and 0.09, respectively. Table A.2 compares base models with instruction-tuned models. Instruction-tuned Llama models (1B, 3B) achieve macro-F1 0.04–0.05 higher than the corresponding base variants.

**Table A.1:** LoRA vs. full fine-tuning: Performance and UQ capabilities for parameter-efficient (using low-rank adaptation; LoRA) vs. full fine-tuning of generative models (in a generative setting). Macro F1 serves as main predictive metric. Head, medium, tail and extreme tail (ET) classes are assessed separately. J refers to the Jaccard score. PRR is the main metric for assessing uncertainty. We report PRR for the best UQ method (see $U_{\text{info}}$ in 4.3). $\rho$ refers to the Spearman correlation coefficient.

| Model | Macro F1 ↑ | | | | | J ↑ | PRR ↑ | $\rho \downarrow$ |
|---|---|---|---|---|---|---|---|---|
| | Overall | Head | Medium | Tail | ET | | | |
| Number of classes → | 1154 | 144 | 481 | 348 | 181 | | | |
| **LoRA fine-tuning** | | | | | | | | |
| Ettin-400M-Decoder | 0.32 | 0.59 | 0.41 | 0.24 | 0.005 | 0.48 | 0.57 | -0.45 |
| Ettin-1B-Decoder | 0.38 | 0.63 | 0.48 | 0.33 | 0.014 | 0.52 | 0.58 | -0.44 |
| Llama-3.2-1B-Instruct | 0.45 | 0.66 | 0.54 | 0.44 | 0.08 | 0.56 | **0.60** | -0.45 |
| Llama-3.2-3B-Instruct | **0.48** | **0.67** | **0.57** | **0.47** | **0.11** | 0.58 | 0.58 | **-0.46** |
| **Full fine-tuning** | | | | | | | | |
| Ettin-400M-Decoder | 0.44 | 0.66 | 0.54 | 0.42 | 0.07 | 0.55 | 0.56 | -0.44 |
| Ettin-1B-Decoder | 0.47 | 0.67 | 0.56 | 0.46 | 0.10 | 0.57 | 0.56 | -0.43 |
| Llama-3.2-1B-Instruct | 0.47 | 0.67 | 0.56 | 0.45 | 0.12 | 0.57 | 0.58 | -0.43 |
| Llama-3.2-3B-Instruct | **0.49** | **0.68** | **0.57** | **0.48** | **0.14** | **0.58** | 0.59 | -0.45 |

**Table A.2:** Base vs. Instruct: Performance and UQ capabilities of base models vs. instruction-tuned models, both fine-tuned in a generative setting. Columns named as in Table A.1 above.

| Model | Macro F1 ↑ | | | | | J ↑ | PRR ↑ | $\rho \downarrow$ |
|---|---|---|---|---|---|---|---|---|
| | Overall | Head | Medium | Tail | ET | | | |
| Number of classes → | 1154 | 144 | 481 | 348 | 181 | | | |
| **Base model** | | | | | | | | |
| Llama-3.2-1B-Base | 0.43 | 0.63 | 0.52 | 0.39 | 0.10 | 0.45 | 0.53 | -0.44 |
| Llama-3.2-3B-Base | **0.48** | **0.67** | **0.57** | **0.46** | **0.12** | **0.58** | **0.70** | **-0.46** |
| **Instruct model** | | | | | | | | |
| Llama-3.2-1B-Instruct | 0.47 | 0.67 | 0.56 | 0.45 | 0.12 | 0.57 | 0.58 | -0.43 |
| Llama-3.2-3B-Instruct | **0.49** | **0.68** | **0.57** | **0.48** | **0.14** | **0.58** | 0.59 | -0.45 |

## A.5 PROMPTING SETUP

### A.5.1 SYSTEM PROMPT

```
SYSTEM_PROMPT = """
You are an AI assistant tasked with classifying a medical device adverse
    event report into one or more categories according to the FDA
    taxonomy. Your goal is to assign all relevant labels to the given
    report.

The report that needs classifying is provided within the <classification-
    text> tag. Along with the report, the label definitions are provided
    within the <labels> tag. To assist you with the task, we also include
     10 "few-shot" examples in the <few-shot-examples> tag. These are
    past reports similar to the one you are classifying – the past
    reports are accompanied by their corresponding labels which were
    tagged by a human expert.

# RULES:

1. The taxonomy of labels is provided within the <labels> tag.
    – Labels are separated by newlines; a definition for the label is
       provided.
    – We are in a 3-level hierarchical multi-label classification setting
        – this means that when a child label (such as A040507) is
       selected, the parent label (A0405) and grandparent label (A04)
       must also be selected. Similarly, if a parent label (A0405) is
       selected, the grandparent label (A04) must also be selected.
    – The converse is not always true – selecting a parent (A0405) or
       grandparent (A04) doesn't necessarily mean selecting all its
       children (A040507).
    – Labels that start with "A" are Medical Device Problems.
    – Labels that start with "E" are Health Effects – Clinical Signs and
       Symptoms or Conditions.
    – The grandparent label (eg. A01) is the most general label, the
       parent label (eg. A0101) is the next most specific, and the child
        label (eg. A010101) is the most specific.
    – The grandparent label always has the letter "A" or "E" followed by
        2 numbers, eg. A01, E01, A02, E02, etc.
    – The parent label always has the letter "A" or "E" followed by 4
       numbers, eg. A0101, E0101, A0201, E0201, etc.
    – The child label always has the letter "A" or "E" followed by 6
       numbers, eg. A010101, E010101, A020101, E020101, etc.

2. There are 10 "few-shot" examples included in the <few-shot-examples>
    tag.
    – Each example includes a report and its corresponding labels.
    – The examples included in the <few-shot-examples> tag were chosen
       using a K-Nearest Neighbours algorithm which picked reports
       similar in content to the text which needs classifying. The
       labels shown for these examples may or may not overlap with the
       labels for the report inside the <classification-text> tag. Use
       them as contextual guidance.

3. Your goal is to classify the text provided within the <classification-
    text> tag.
    – Assign all labels that are relevant.
    – You can choose multiple labels, a single label, or no labels if
       none apply.
    – Always use the exact label names from the label list provided in
       the taxonomy under the <labels> tag.  Do not invent new labels or
        modify existing ones.
    – Return your output as a list of labels, separated by newlines.
    – Do not include any explanations, text, or formatting outside the
       label list.
```

```
1134        - If no label applies, return an empty list.
1135        - Do not invent new labels.
1136
1137   4. Provide your output as a list of labels, each on a new line. For
1138        example:
1139
1140   A04
1141   A0405
1142   A040507
1143   E01
1144   E0101
1145
1146   # IMPORTANT
1147   *In your final output, you must not include any extra text, explanations,
1148        or formatting outside the label list. Only return the list of labels
1149        separated by newlines.*
1150   """
```

## A.5.2   USER PROMPT

```
USER_PROMPT = """
You are an AI assistant tasked with classifying a medical device adverse
    event report into one or more categories according to the FDA
    taxonomy. Your goal is to assign all relevant labels to the given
    report. The rules were provided in the system prompt. For the sake of
    clarity, we are repeating them here:
- Assign all the labels that are relevant to the report, but only if you
    are sure about it.
- You can choose multiple labels, a single label, or no labels if none
    apply.
- Use exact label names from the provided taxonomy. Do not invent or
    modify labels.
- Include parent and grandparent labels when selecting a child label.
- Do not include any explanations, text, or formatting outside the label
    list.
- If no labels apply, return an empty list.
- Your output should contain only the list of applicable labels, with
    each label on a new line. You must not include any extra text,
    explanations, or formatting outside the label list.
- Provide your output as a list of labels, each on a new line.
Example output:
A04
A0405
A040507
E01
E0101

Here's how to proceed:

1. First,  familiarize yourself with the label definitions:

<labels>
A01: Patient Device Interaction Problem - Problem related to the
    interaction between the patient and the device.
A0101: Patient-Device Incompatibility - Problem associated with the
    interaction between the patient's physiology or anatomy and the
    device that affects the patient and/or the device.
.
.

</labels>

2. Review these few-shot examples of similar reports and their
    corresponding labels:
```

```
<few-shot-examples>
{EXAMPLES}
</few-shot-examples>

3. Now, carefully classify the following report:

<classification-text>
{CLASSIFICATION_TEXT}
</classification-text>"""
```

