# OpenReview forum: "From small to large language models: How much confidence can we have?"
_ICLR.cc/2026/Conference — ICLR 2026 Conference Withdrawn Submission_

### Official Review · Reviewer_DefB · 2025-10-28

**Soundness:** 2
**Presentation:** 3
**Contribution:** 2
**Rating:** 2
**Confidence:** 4

**Summary:**

This work quantifies three types of uncertainty (into information-based, consistency-based and self-verbalized) of 20+ encoder- and decoder-only language models in three context: discriminative fine-tuning, generative fine-tuning, and few-shot in-context prompting. Experimental results were analyzed to provide practical guidance.

**Strengths:**

* Use multiple uncertainty quantification methods
* Evaluate 20+ models.

**Weaknesses:**

* How to ensure there is no pre-training exposure? The proposed dataset was compiled from information publicly available at  https://open.fda.gov, which may have been used to pretrain LLMs
* Lines 112-113: Softmax is used. Are those labels mutually exclusive?
* This work claims that it "systematically study multiple UQ approaches ... including information-, consistency- and self-verbalized based uncertainty". Please discuss why chose them and How they overlap and differ. Note: Harsha Tanneru et al. (2024) proposed "verbalized uncertainty" and reported that "verbalized uncertainty is not a reliable estimate of explanation confidence".

**Questions:**

* Line 275: Are U_info and U_cons independent? What does their multiplication mean semantically?
* The "self-verbalized uncertainty", explained in lines 294-295, may not be the uncertainty of the prediction value produced by model.  How does an LLM generate a "confidence score"? Do the "confidence scores" produced by all LLMs mean the same thing?

---

### Official Review · Reviewer_kUyz · 2025-10-31

**Soundness:** 2
**Presentation:** 3
**Contribution:** 2
**Rating:** 4
**Confidence:** 4

**Summary:**

The paper examines uncertainty quantification for multi-label text classification using over 20 recent language models in three usage setups: discriminative fine-tuning, generative fine-tuning, and few-shot prompting. It introduces a large dataset of FDA medical device reports labeled with hierarchical codes. The study compares information-based, consistency-based, and self-verbalized confidence methods. It finds that generative models provide the most reliable uncertainty estimates, while self-verbalized confidence performs poorly.

**Strengths:**

1. This paper introduces a large labeled dataset of FDA medical device reports, which might not overlap with the existing dataset, allowing for a fair evaluation.
2. The experiments are done over three different settings and over 20 models.

**Weaknesses:**

1. The writing of this paper lacks some important details. For example, there is no information about which U_{info} is used in the Table1. Besides, for consistency-based methods and reasoning models, whether CoT is considered in the consistency calculation is also not mentioned.
2. The evaluation is only based on the text classification task, which constrains the conclusion a lot since many methods are designed for open-end generations like the consistency used in this paper.  I also doubt some choices of U_{info} used in the paper. For example, Perplexity and Avg(\pi) should be exactly the same for ranking ability since exp is a monotonous function and will not influence the ranking. And the evaluation metric PRR only considers the rank ability.
3. Lack of some important methods in UQ. For example, p(true) [1] should be considered as a self-verbalized uncertainty and it performs much better than asking models to output a confidence. Besides, I am also wondering how semantic entropy [2], which combines both consistency and information methods, can be categorized in this paper’s framework. Besides, it seems less meaningful to use U_{cons} in the current setting because U_{cons} is based on the semantic similarity, that is why the original U_{cons} paper shows that using a natural language inference model could get a better result. However, the current output from models (A0102) does not contain any semantic information.
4. The evaluation of ‘calibrated’ UQ seems not accurate in the paper. Normally, calibration of UQ refers to expected calibration error (ECE) [3] instead of the ranking ability of UQ.
5. The findings from the experiments lack of in-depth explanations. For example, the author claims that ‘generative fine-tuning yields the strongest UQ, followed by discriminative fine-tuning’, which is a good start to a series of important investigations. However, the authors do not provide analysis on why generative fine-tuning can yield the strongest UQ. Which difference between generative and discriminative fine-tuning causes the different performances in UQ? Why is generative fine-tuning better? Is it because of the training prompt? The additional classification layer or something else? Then why does Llama-3.2-3B-Base and Llama-3.2-8B-Base have a better PRR for discriminative fine-tuning instead of generative fine-tuning?  Then for reasoning, what cause the reasoning models to perform worse in UQ?


[1] Kadavath, Saurav, et al. "Language models (mostly) know what they know." arXiv preprint arXiv:2207.05221 (2022).
[2] Kuhn, Lorenz, Yarin Gal, and Sebastian Farquhar. "Semantic Uncertainty: Linguistic Invariances for Uncertainty Estimation in Natural Language Generation." The Eleventh International Conference on Learning Representations.
[3] Guo, Chuan, et al. "On calibration of modern neural networks." International conference on machine learning. PMLR, 2017.

**Questions:**

Please see the weakness part.

---

### Official Review · Reviewer_km3S · 2025-11-09

**Soundness:** 3
**Presentation:** 3
**Contribution:** 1
**Rating:** 2
**Confidence:** 4

**Summary:**

The paper evaluates different LLMs on a self-curated biomedical dataset to aim to understand when to select an appropriate approach for a given model. The evaluation is extensive, using accuracy and UQ metrics, with particular attention paid to the latter. The evaluation is also done across models after discriminative finetuning, generative finetuning, and reasoning prompting. The paper finds that generative finetuning provides the most reliable UQ, smaller encoder models achieve competitive performance after discriminative finetuning, and thinking models have weaker UQ.

**Strengths:**

The strengths of this paper are the scope of experiments and clarity. The paper cross-compares three areas. Each area is important and the research community would be interested in learning how it influences the LLM: 1) different ways of strengthening a model (namely, two finetuning methods and one in-context/prompting method), 2) model types and sizes; 3) their UQ capabilities. The writing is pretty clear as well.

The paper also curates a dataset that would help the LLM community to study and improve LLMs.

**Weaknesses:**

A weakness of the paper is a lack of central claim or central finding. While experiments cross-compare different aspects of language models, the results are hard to interpret. The paper summarizes several patterns from the results (such as those I mentioned in 'Summary'), but I cannot find a deep analysis of these patterns.

The current experiments are a good start, but I would suggest that the paper finds one or two of these result patterns and does a closer study. Just as an example, the paper finds that reasoning models offer weaker UQ. It is possible to look into why they are weaker in UQ and use additional experiments to test these claims. Then, the phenomenon of reasoning models offering weaker UQ will be more convincing.

Another aspect I would like to know more about is why generative finetuning has the best UQ, and in particular better than discriminative UQ. Intuitively generative finetuning is harder for UQ because there is semantic uncertainty vs token uncertainty: there are different words that can achieve the same meaning (e.g. "I'm happy to" vs "I'm glad to"). More analysis on the nature of these two types of finetuning would help. Currently, the picture is still unclear to me.

**Questions:**

Additional suggestions:

* Abstract - "We provide practical guidance on model selection, when fine-tuning is preferable to prompting, and which UQ signals are most effective for routing and human-in-the-loop triage" - grammatical error, maybe needs a 'such as' before 'when'
* Introduction - Needs a paragraph break in the long first paragraph
* Additional citation suggestion in UQ: Gruver, et al. (2023) - Large Language Models Are Zero-Shot Time Series Forecasters

---

### Official Review · Reviewer_Havv · 2025-11-12

**Soundness:** 2
**Presentation:** 2
**Contribution:** 2
**Rating:** 2
**Confidence:** 4

**Summary:**

The paper examines uncertainty quantification (UQ) for multi-label text classification across various language models to provide guidance for model selection. The authors propose a new unsaturated benchmark (to avoid training data leakage) of medical device adverse event reports with a hierarchical and multi-label structure. The authors fine-tune or prompt over twenty language models ranging from small encoder-only to large decoder-only models, examining different learning paradigms: discriminative fine-tuning, generative fine-tuning, and few-shot prompting.

**Strengths:**

**The following are the strengths of the paper:**
1. This paper compares different uncertainty quantification (UQ) measures for multi-label text classification over 20 language models.

2. The authors create a new dataset of medical device adverse event reports with a hierarchical and multi-label structure based on FDA and IMDRF terminology.

**Weaknesses:**

**The following are the weaknesses of the paper:**
1. The primary contribution of this paper is introducing a new dataset designed for uncertainty quantification (UQ) of different language models.

2. The paper does not propose any novel problem formulation, method, or new UQ metric.

3. As the evaluation depends solely on a single dataset, the findings and insights presented in the paper may be biased or limited in their generalizability.

**Questions:**

I have a few more questions/comments:

**Details Of Ethics Concerns:**

I do not find any ethical concerns.

---

### Note · Authors · 2025-11-18

I have read and agree with the venue's withdrawal policy on behalf of myself and my co-authors.